# Structural Interventions to Enable Adolescent Contraceptive Use in LMICs: A Mid-Range Theory to Support Intervention Development and Evaluation

**DOI:** 10.3390/ijerph192114414

**Published:** 2022-11-03

**Authors:** Helen Elizabeth Denise Burchett, Sally Griffin, Málica de Melo, Joelma Joaquim Picardo, Dylan Kneale, Rebecca S. French

**Affiliations:** 1Department of Public Health, Environments & Society, Faculty of Public Health & Policy, London School of Hygiene & Tropical Medicine, London WC1H 9SH, UK; 2International Center for Reproductive Health, Maputo 1100, Mozambique; 3EPPI-Centre, UCL Social Research Institute, University College London, London WC1H 0NR, UK

**Keywords:** contraception, family planning, adolescent, structural, upstream, intervention evaluation, cash transfer, schooling, norms, empowerment

## Abstract

Enabling contraceptive use is critical for addressing high adolescent pregnancy rates in low- and middle-income countries (LMICs). Broader or ‘upstream’ determinants, such as poverty, education, and social norms, can affect the knowledge, attitudes, motivation, and ability to access and use contraception. Structural interventions aim to address these broader determinants, e.g., through poverty alleviation from livelihood training or cash transfers, increasing school participation, or changing social norms. We conducted an evidence synthesis using intervention component analysis, a case-based approach, following a systematic mapping of the evidence base. We identified 17 studies with 29 structural intervention arms, which reported adolescent contraceptive use outcomes compared to a control group or baseline. It was not possible to identify with certainty which interventions were ‘likely effective’ or ‘likely ineffective’ due to the high heterogeneity of the methods. We built on an existing framework of family planning use to propose three steps to designing interventions: (1) tailor interventions to adolescents’ life stages; (2) assess the baseline situation; and (3) select appropriate activities to match the gaps. These steps will aid developers and evaluators of structural adolescent contraceptive interventions to develop an evidence base that is of use across a wide range of settings and use scenarios.

## 1. Introduction

Despite progress in some regions, adolescent pregnancy rates remain high in many low- and middle-income countries (LMICs) [1]. Contraceptive use is an important means of avoiding, spacing, or delaying childbearing. The prevalence of contraceptive use is typically lower among sexually active adolescent girls and young women (hereafter referred to as ‘girls’) in LMICs compared to older women [2]. Most of the evidence on the effectiveness of adolescent contraceptive interventions has focused on supply-side factors and/or targeting girls with information and education in order to change their knowledge, attitudes, and contraceptive behaviours [3,4,5]. However, a range of broader determinants, such as poverty, education and social norms related to gender, sexual behaviour, and fertility, affect not only their knowledge, attitudes, and motivation to prevent pregnancy but also their ability to access and use effective contraceptive methods [6,7]. Structural interventions aim to address these broader determinants by changing the structural context within which sexual and reproductive health behaviours take place, for example, by reducing poverty, enabling girls’ participation in education and the workforce, empowering girls, and changing social norms around gender equity, sexual behaviour, and fertility. They target the “contextual or environmental factors that influence risk behavior…rather than in characteristics of individuals who engage in risk behavior” ([8] p. 59). In contrast, non-structural interventions to increase contraceptive use may focus on increasing access to and information on contraceptives and tend to assume high levels of autonomy and agency on the part of adolescent girls. Consequently, these interventions may not address the social causation of early fertility or strengthen autonomy and agency among adolescent girls.

The mid-range theory could help those commissioning, developing, or delivering adolescent contraceptive interventions in specific contexts through the integration of broad theories (in this case around reproductive decision making and contraceptive use) with empirical evidence from intervention evaluations. Previous adolescent reproductive health syntheses have noted a lack of theory underpinning the interventions or the need to document why or how the interventions work [9,10,11].

Prior to the current study, we conducted a systematic map of the literature [12] In the current study, we conducted an evidence synthesis to identify the characteristics of structural adolescent contraceptive interventions in LMICs that may facilitate or hinder their effectiveness. We also aimed to develop a mid-range theory to explain how structural interventions work to enable effective contraceptive use among adolescents in LMICs.

## 2. Materials and Methods

Prior to the current study, we conducted a systematic map of the literature (see [12] for details of our search strategy and inclusion criteria). Studies were included in this map if they were published in 2005 or later, were conducted in a LMIC, were an intervention evaluation reporting contraceptive or pregnancy/birth outcomes, focused on or reported outcomes for adolescents (10-19 years) and if the intervention was upstream/structural. For the current study, from those studies included in the map, we included studies if they reported outcomes for adolescents relating to contraceptive use (as defined by study authors) that were compared either with a (non-historical) control group or with baseline data.

We extracted contraceptive-use outcomes, selecting the current use of hormonal/barrier contraceptives at the 12-month follow-up where possible or else at the timepoint nearest to 12 months. We examined the distribution of the effect sizes using meta-analytic methods to enable us to categorise studies as ‘likely effective’, ‘possibly effective’, ‘possibly ineffective’, or ‘likely ineffective’ (see Table 1). These categories reflected a balance of the magnitude (size) and direction of the effect size as well as its precision. Those studies where an effect size could not be calculated or where the precision of an effect size could not be determined were manually assigned to a category.

Due to the variety of study designs and outcome measures, although it was possible to harmonise the effect sizes to the same effect size unit through a number of transformations (e.g., using Chinn’s formulae for converting effect sizes and standard errors between standardised mean differences and odds ratios [13]), it was not possible or appropriate to meta-analyse the data to create pooled effect sizes. Although the effect sizes were included in the same forest plot, this was for illustrative purposes to help identify the likely effective and ineffective studies.

We then focused on the ‘likely effective’ and ‘likely ineffective’ studies to explore what could explain the difference between these two sets. This is a method that has previously been used in qualitative comparative analysis (QCA) and allows a focus on the differences between these two groups, avoiding the ‘noise’ from those achieving a moderate effect [14]. This is important since heterogeneity is of critical importance in QCA in order to identify which combinations of characteristics are sufficient to explain the outcomes [15].

Our study differed from the planned methods as set out in our protocol in two ways [16,17]. Firstly, we had planned to quality appraise the studies, excluding those of a low quality and small sample size, before conducting an intervention component analysis (ICA) in order to inform the QCA (which would explore intervention characteristics in terms of their content and implementation and the context within which they took place). However, it became apparent that methodological issues undermined our confidence in the comparability of the contraceptive use findings (see the Results section). Studies used a variety of measures and many studies reported contraceptive use as a secondary outcome, with disparate and/or small samples (such that excluding those would leave few remaining in the synthesis). Due to this high heterogeneity and since the methodological issues noted were not captured by the quality appraisal tools, we conducted a QCA of the methodological factors instead of quality appraising the studies.

Secondly, given the methodological issues identified, it became meaningless to focus on trying to understand the differences between the ‘likely effective’ and ‘likely ineffective’ sets of studies as we were not able to say definitively what did or did not work. Instead of the planned QCA of the intervention and contextual factors to develop a mid-range theory, we focused on using ICA as a case-based approach to understanding what happened within each study.

ICA is an iterative, case-based approach that brings evidence from all parts of an intervention evaluation report to develop a theory [18]. Firstly, the intervention reports were read several times before the information was extracted about the intervention arms’ contexts, contents, and implementations, as well as the evaluation designs, process evaluation findings, authors’ explanations, and intermediate indicators. We considered the interventions’ characteristics, the characteristics of the evaluations and authors’ insights into why they believed their interventions were or were not effective, and the insights into the implementations and contexts of the studies. We compared and contrasted the studies and, where different intervention arms within a study yielded different results, we explored the possible factors that could explain this variance. Discussions among the team were held where we considered the different aspects of the interventions and their evaluations and how they might affect contraceptive use. When a potentially important intervention element was identified, the full set of studies was then reviewed with this element in mind to explore whether there was evidence to support or refute it. This iterative process continued until the theory presented below had been developed.

## 3. Results

We screened 6993 references and included 29 arms from 17 studies. Ten studies were conducted in Africa, four in Asia, and three in South America. Structural intervention activities included those aiming to develop economic empowerment, encourage adolescent girls’ school participation, and actively work with communities to change social norms around gender and adolescent sexuality/fertility (see Appendix A Table A1 for details of studies; or Burchett et al. for further discussion of structural interventions).

We categorised five study arms as ‘likely effective’ in terms of increasing contraceptive use and five as ‘likely ineffective or harmful’ (hereafter ‘likely ineffective’); one study, *DISHA*, was assigned to the ‘likely effective’ category although it was not included in the meta-analysis as it did not report control arm data, only baseline. The remaining study arms were considered ‘possibly effective’ or ‘possibly ineffective’ (see Figure 1).

### 3.1. General Methodological Characteristics of Included Studies

The included interventions had a range of aims including reducing poverty, preventing HIV, delaying child marriage and/or delaying sexual initiation, empowering young women to reduce unintended pregnancy, and improving sexual and reproductive health outcomes (see Appendix A Table A1). Most either implicitly or explicitly aimed to increase contraceptive use (*n* = 13/17); however, four had other aims (HIV prevention [19,20]; delaying marriage [21]; or poverty reduction [22]).

The heterogeneity of several methodological characteristics, such as the study design, outcome measure used, the outcome measure sample, timing of the follow-up, and confounders, made categorising the included studies as ‘likely effective’ or ‘likely ineffective’ challenging (see Appendix A Table A2 and Table A3).

Seven studies used a randomised control trial (RCT) study design; six of these used a cluster-randomised controlled trial (cRCT) design and two were individually randomised^4^ (one study, *AGI-K*, used a cRCT design in one arm and an individually-randomised controlled arm in the other; another study, *CERCA*, used an RCT design in one arm and a non-RCT design in two other arms) (see Appendix A Table A2 for details of the studies’ designs and outcomes). Eight studies used non-randomised (quasi-experimental) designs, two were natural experiments, and one used a pre–post-intervention design.

The control groups varied in terms of the intervention they received or the extent to which they may have been contaminated with the intervention. For example, the control groups for the *Adolescent Girls Initiative—Kenya (AGI-K)* and *Shaping the Health of Adolescents in Zimbabwe (SHAZ!)* studies both received substantial interventions; in the former, community conversations were held on violence prevention and valuing girls [23] and in the latter, life skills training and free contraceptives were offered [20]. Five studies had no baseline; in these, the effectiveness was assessed through comparison to a control group at endline [19,22,24,25,26]. One study, *Development Initiative Supporting Healthy Adolescents (DISHA)*, had no control group; in this, they compared outcomes pre- and post-intervention [27].

Where available (*n* = 7), we used the outcome ‘current contraceptive use’ at the 12-month follow-up (see Appendix A Table A2). In the remainder, other measures of contraceptive use were included, e.g., ‘ever use of contraception’ or ‘current use’ at a different time point (e.g., 3–4 years later) and ‘contraceptive use at last sex’ or ‘ever used contraception to delay their first birth’. Differences in the timing of the outcome data collection could be critical, not only for those measuring ‘ever use’ but also those with different age groups (e.g., younger women are less likely to be sexually active and therefore less likely to need contraception, but also sexually active younger women are less likely to use contraception than older women [28,29]. With a longer time period, the study samples become older, affecting the likelihood of contraceptive use regardless of the intervention effects).

All but one study used self-reporting methods to capture this outcome; the exception (*Girl Power—Malawi*) used clinic data for the receipt of hormonal implants, injections, or the contraceptive pill, corroborated with self-reported use [30]. There was also variation in what types of contraception were included in the outcome. Some studies, e.g., *Promoting Change in Reproductive Behaviour of Adolescents (PRACHAR III)*, included emergency contraception [26], whereas others asked about condom use separately from hormonal contraception, which may have led to underestimates of actual contraceptive use (e.g., *Empowerment & Livelihoods for Adolescents—Uganda (ELA Uganda); AGI-K*) [23,31]. Several did not specify which contraceptive methods were included in their outcome measures.

Contraceptive use was measured among different samples (e.g., in some studies, those who received the intervention were then followed up at endline, whereas others sampled adolescents from the intervention community regardless of whether they had been directly exposed to the intervention). Some only asked married respondents about their contraceptive use (e.g., in *Berhane Hewan* [32]). One study, *Community-embedded Reproductive Health Care for Adolescents (CERCA)* asked all endline participants without disaggregating male/female responses and counted all those who had never had sex as not using contraception [33]. Only one study, *SHAZ!*, specified that they asked this question of those who had reported having sex in the previous month [20]; two others reported only asking those who were ‘sexually active’ but failed to define whether this meant that they had ever had sex, had had sex within a specific period of time, or whether the sex was vaginal [31,34]. In general, studies’ questions about sex did not distinguish between vaginal and other types despite this having implications for contraceptive outcome measures.

Several interventions aimed to delay sexual initiation and, when successful, the intervention itself would therefore have reduced the size of the subsample of participants who had ever had sex and were asked about their contraceptive use relative to the control arm. The conceptual issues in the changing nature of the denominator were not often acknowledged within the trial reports. In some studies, only a minority of participants had ever had sex at endline [23,33] or it was unclear how many had ever had sex at endline (e.g., *PRACHAR III; DISHA*) [26,27]. It should also be noted that despite most studies only asking participants who had ever had sex about their contraceptive use, this assumed that those who had ever had sex were continuing to have regular sex. However, data from the *SHAZ!* study showed that only a minority of those who had ever had sex had done so in the previous three months [20]. Even among a group of married and/or parenting adolescents in the *Gender Roles, Equality and Transformations project (GREAT)* study, only 70–80% had had sex in the previous three months [35]. This has implications for ‘current contraceptive use’ as an outcome measure; sexually initiated respondents who were not currently sexually active would be recorded as not using any current contraception despite not being at risk of pregnancy. Arguably a more pertinent outcome measure would be the frequency of unprotected vaginal sex; however, this was only rarely used (e.g., *ELA—Sierra Leone*) [34].

### 3.2. Methodological Characteristics of the ‘Likely Effective’ and ‘Likely Ineffective/Harmful’ Study Arms

None of the studies with arms categorised as ‘likely effective’ had been evaluated with an RCT study design, whereas two of the ‘likely ineffective’ had been evaluated with an RCT study design (*SHAZ!* and *CERCA*) [20,33]. The QCA identified methodological issues in both the likely effective and likely ineffective sets of studies as well as the possibly effective and possibly ineffective sets. A table of the methodological characteristics of the studies is shown below (see Table 2).

Using QCA methods, we developed a truth table from this data table with some of the characteristics that might detract or improve confidence in the findings the most—whether the study was an RCT, whether it included a control group, and whether the measures were collected among a relevant population (sexually active females) (Table 3). We were only able to identify a configuration with a single likely effective study with methodologically weak characteristics, with other likely effective studies distributed across other configurations.

Notably, the truth table shows that no configuration of studies was identified that had no risk of bias in terms of selection (offset by an RCT design) and no risk of bias in attribution bias (offset by having a control group and measuring the outcome among sexually active participants). In short, it is not possible to say with any certainty which characteristics of the interventions, their implementations, or the settings in which they were evaluated were associated with ‘likely effective’ or ‘likely ineffective’ studies; any observations made would be undermined by serious methodological weaknesses. Instead, the remainder of the paper presents an alternative synthesis approach using intervention component analysis.

### 3.3. A Mid-Range Theory for Contraceptive Use Interventions

We used ICA to develop a mid-range theory using evidence from the full set of included studies and not just those identified as ‘likely effective’ or ‘likely ineffective’. To develop a mid-range theory that can then be tested through rigorous evaluations in the future, we used existing conceptual models and frameworks as our foundation. This meant that we were not ‘reinventing the wheel’ but building on cumulative knowledge in combination with examples from the intervention evaluations identified. We built on an existing conceptual framework developed by the International Center for Research on Women (ICRW) [9]. Their framework contains three demand-side and two supply-side objectives (plus an enabling environment—hereafter referred to as the sixth objective) that they propose leads to sustained effective contraceptive use (see Figure 2).

Although objectives 2, 4, and 5 could be addressed through non-structural interventions (e.g., information provision to increase the desire to use contraception and service delivery improvements to increase access to contraceptive services and provide quality, youth-friendly services, hereafter ‘YFS’), objectives 1, 3, and 6 (desire to avoid/delay/space/limit childbearing, agency to use contraception, and an enabling environment) are strongly influenced by upstream factors that are likely to be best addressed by structural interventions. We, therefore, theorise that for contraceptive use to be enabled, all six objectives need to be met. Some objectives may already have been met at baseline, in which case, the interventions should focus on the outstanding objectives. In many cases, interventions will need to be multi-component, incorporating both structural and non-structural elements in order to ensure all six objectives are met.

We propose three steps for designing effective interventions that ensure that the six objectives are met based on the ICA analysis of the studies included in this review: (1) tailor interventions to the adolescents’ life stages; (2) assess the baseline situation for each objective; and (3) select appropriate intervention activities to match the objectives.

#### 3.3.1. Step 1: Tailor Interventions to the Adolescents’ Life Stages

Although all of the interventions targeted adolescents, only five took into account different life stages. One focused exclusively on married adolescents, albeit not distinguishing outcomes between nulliparous girls and parents [36], one focused on very young adolescents [23], and three provided different interventions depending on life stage: for unmarried and married girls [32], for unmarried nulliparous and married/parents (indeed this intervention explicitly aimed to “test life-stage specific strategies” p. 1) [35], or tailored the interventions by both marital status and parity [26].

Adolescence is a time when girls can transition rapidly through different life stages (e.g., nulliparous or parents; married or unmarried, with or without a regular partner), and will likely have very different situations, needs, and intervention requirements relating to contraception. Those commissioning or developing the interventions and those designing the evaluations must recognise that adolescent girls are not a homogenous group and should decide explicitly which life course stage(s) they wish to target before the intervention activities are selected and to ensure the evaluations assess effectiveness in different subgroups.

Married adolescent girls often experience different pressures and experiences than unmarried girls. Some studies noted the social pressure, often exerted by family and partners as well as the wider society, for newly married adolescent girls to have children, e.g., *PRACHAR III* [26]. Yet at the same time, contraception may also become more accessible as it becomes more socially acceptable for married women to be sexually active, e.g., *GREAT* [35]. Some interventions noted that it is easier to increase contraceptive use among married women compared to unmarried, e.g., *Regai Dzive Shiri* [19], although others were able to increase use among both married and unmarried women (albeit with higher rates among the former), e.g., *Girl Power—Malawi* [30]. The importance of tailoring interventions to married adolescent girls at different life stages was noted by those evaluating the *First Time Parents Project*, who recognised that some would be trying to conceive, whereas others would want to delay their first pregnancy or would already be pregnant or new mothers [36].

Motherhood is a life stage that can affect the ease with which adolescent girls feel willing and able to use contraception. Evidence from the included studies shows that interventions are often more successful at spacing subsequent births than avoiding first births [26,37]. Social and familial pressures to have children can ease after the first birth and new parents can gain access to health services that may have been harder to access when nulliparous.

Even among unmarried nulliparous girls, circumstances may vary. Those with a regular sexual partner are more likely to use a hormonal contraceptive, whereas those without are less likely to use contraception overall but if they do, they are often more likely to use condoms [35,38].

Younger adolescents and those in school are often easier to reach than older adolescents and those out of school—which could affect not only what interventions are suitable and how best to recruit and engage participants but also the outcomes that the evaluations achieve [21,33,35]. How livelihood training or support for schooling is experienced and the effects that they have may differ depending on the life course stage and age of the adolescent girls involved. Some studies found a greater impact among older adolescents compared to younger, possibly due to greater agency or reflecting the differences between those having sex at a younger age and those with a later sexual debut [29,34].

Not only could the baseline situation in relation to each objective of the ICRW framework vary by life stage but also the activities required for each objective may vary. The framework could be further broken down by life stage, for example, the desire to space births among mothers is distinct from the desire to delay or avoid childbearing among nulliparous women; the same intervention activities may not have the same effect on both of these ‘sub-objectives’. However, spousal communication and spousal support for contraception may be important to increase girls’ agency to use contraceptives among married adolescents (as will be discussed in more detail later); for unmarried adolescents without a regular partner, this is less likely to have an immediate effect but rather may be of use in the future when they have a regular partner.

#### 3.3.2. Step 2: Assess the Baseline Situation for Each Objective

It seems logical that all six objectives would need to be met in order to attain higher rates of contraceptive use. However, it may be that the interventions do not necessarily need to target all six objectives if one or more have already been met.

Although most studies had some form of baseline (although not all used these to compare outcomes at endline) or formative research, few reported the baseline situation for the six objectives. Furthermore, given the lack of consensus about which indicators are most appropriate to measure each objective, it is not possible to compare the studies’ baseline situations.

In the first step of the intervention development process, the contexts and experiences of adolescents (at the life stage being targeted) should be assessed to ascertain the baseline situation in relation to each objective. This will allow an understanding of which objectives should be focused on and prioritised in the intervention package. For example, married adolescent mothers may already have a desire to space or limit childbearing but may lack the desire or agency to use family planning. In this case, an intervention should target these two objectives rather than the former. In assessing the context, the need for tailored consideration remains. For example, it may be that youth-friendly contraceptive services exist but are only ‘youth friendly’ for married youth, with barriers or concerns about confidentiality still perceived by unmarried adolescent girls.

Understanding the local context is of critical importance. For example, if there is already a high desire to avoid or delay childbearing among unmarried adolescents with no regular partner, interventions need not focus on activities to increase it, as was the case among unmarried adolescents at baseline in *PRACHAR III* [26]; therefore, intervention efforts should focus on other parts of the pathway. In summary, interventions may not need to address all six objectives but rather, an understanding of the baseline situation is required.

#### 3.3.3. Step 3: Select Appropriate Intervention Activities to Match Objectives

Studies rarely stated explicitly whether they were attempting to address specific objectives within the broader goal of increasing contraceptive uptake (e.g., whether they aimed to increase the desire to limit, avoid or space births, or improve access to family planning services).

Once the target subpopulation has been selected and the focus of the intervention has been determined, specific intervention strategies can be selected. Structural intervention activities could most usefully target three objectives in the framework: objective 1/desire to avoid, delay, space, or limit childbearing, objective 3/agency to use contraception, and objective 6/an enabling environment. The remaining three objectives could be addressed primarily through non-structural interventions (e.g., mass media campaigns or sex education for objective 2/ desire to use family planning; service delivery improvements for objectives 4/ access to family planning and 5/ YFS). However, structural interventions could still have a direct or indirect effect on these, for example, an intervention aiming to increase participation in school could increase the desire to use family planning by increasing access to school-based sex education, whereas economic empowerment interventions could increase the affordability of contraception, thereby addressing access to methods.

A.Interventions aiming to increase the desire to limit/avoid/space/delay childbearing

Within this objective, we view the desire to limit, avoid, or delay (first) childbearing as distinct from the desire to space births (the ethics and public health benefits of avoiding or delaying childbearing among older adolescents is beyond the scope of this project, but should nevertheless be considered by those developing or funding interventions). Almost all of the studies focused either on delaying or avoiding first births or did not specify. None focused solely on the objective of spacing subsequent births, although a small minority of studies explicitly targeted this alongside delaying first births, e.g., *GREAT*, *PRACHAR III* [26,39]. Interventions aiming to delay or space births often provided information about the risks of early pregnancy and short birth spaces or the benefits to the mother and existing child(ren) of delaying or spacing births, as well as structural interventions.

The structural interventions included in our review, which aimed to increase adolescent girls’ desire to delay or avoid first births, did so by trying to increase girls’ value aside from motherhood. This may relate to either their potential future perceived value (e.g., by increasing their aspirations or education and future employment opportunities) or their current value (e.g., through skills training or economic opportunities). Interventions could target girls’ perceptions of their own value, the perceptions of the husband or family, and/or the wider community.

Interventions to increase girls’ future value aimed to increase their aspirations through vocational training, encouraging schooling, or life skills. For example, the *BALIKA* study included an arm providing vocational training for two weeks, which aimed to increase aspirations (as opposed to providing sufficient training for work) [21]. The *Sawki* study aimed to enhance girls’ current value through income generation activities [25], whereas in *Berhane Hewan*, married adolescent girls were given skills and support to improve the nutritional status and living conditions of their families, through gardening and learning to build basic furniture and more efficient cooking fires [40].

Few studies measured the desire to delay or space births or the indicators of aspirations. An exception was the *Empowerment and Livelihood for Adolescents—Uganda (ELA—Uganda)* study, which aimed to “break the vicious cycle between low participation in skilled jobs and high fertility” ([31] p. 212). They combined vocational training and life skills training, delivered through a safe space model. They found that perceptions of what age is suitable for women to have their first child increased significantly among girls in the intervention arm compared to those in the control arm [31]. The *Sawki* study in Niger compared a control arm to a safe space intervention and a safe space plus livelihood training intervention [25]. Girls in both intervention arms reported a higher ideal age at childbirth compared to the control; this was highest in the arm offering livelihood training.

B. Structural interventions aiming to increase agency to use contraception

The concept of agency is fundamental to many of the included studies, albeit addressed in diverse ways. The ICRW framework describes barriers to the agency to use contraception such as limited decision-making autonomy and power for girls, early marriage, family pressures, poor partner communication, sexual violence and transactional sex, and limited self-efficacy [9]. It is clear that ‘agency’ is a multidimensional construct requiring further unpacking.

A conceptual model of women and girls’ empowerment developed in 2017 recognises three key elements within agency: choice, voice, and power [41]. Although the included studies provided examples of activities that aimed to develop one or more of these three elements, there is insufficient evidence to identify which aspects of agency should be targeted or how to increase specific aspects of agency in relation to contraceptive use.

Intervention activities aiming to increase adolescent girls’ agency targeted at least one of three levels: societal, interpersonal, and the individual adolescent girl.

Activities targeting the societal level, such as gender transformative interventions aiming to shift norms around women and girls’ roles, value, and gender equity, were present in seven studies (although two of these also offered these activities to the control arm). These activities overlap with those for objective 6 (develop an enabling environment) and are discussed further in section C below.

Most included interventions (*n* = 11/17) targeted at the interpersonal level, addressing the role of boys, partners, parents, and/or the wider community. However, when we look at how the studies aimed to increase interpersonal agency, most did so through communication skills training targeting girls or girls and boys. Some also incorporated training or awareness raising of girls and boys, partners and/or parents, or the wider community around gender norms and healthy relationships. Nevertheless, some studies did go beyond training or awareness raising, for example, creating discussion groups for ‘adult-youth partnerships’ (*DISHA*) [27], for parents (*Regai Dzive Shiri; CERCA*) [19,33] and for young married husbands (*PRACHAR III*) [26].

All the included interventions targeted the individual adolescent girl. We identified several different intervention options used in the included studies for each element of van Eerdewijk et al.’s model of choice, voice, and power (see Table 4 [41]).

Assessing whether or not interventions were successful in empowering girls is challenging without a consensus about what empowerment is, particularly in relation to contraceptive use or what indicators could measure it. Although a range of indicators was used to capture the different aspects of agency, it is difficult to know which indicators map directly to contraceptive agency. For example, in *AGI-K*, compared to arm 1 (control), arms 3 and 4 had no effect on general self-efficacy, girls’ perceptions of gender norms, or the acceptability of intimate partner violence. However, there was a positive impact on condom self-efficacy and help-seeking self-efficacy [23]. The extent to which condom self-efficacy and help-seeking self-efficacy align with contraceptive self-efficacy is not clear, particularly since the former requires much greater buy-in from male partners than other, female-controlled contraceptive methods.

The most commonly used agency indicators were related to spousal communication and/or spousal support (or adolescent girls’ perceptions of their support) for contraception. Only a minority of studies evaluated whether the interventions had an effect on partners, parents, or community members, e.g., *CERCA* asked adolescents whether they communicated with their partner or parents about ‘sexuality’, and *DISHA* measured adults’ attitudes to whether contraceptive information should be available for adolescents [27,33].

C. Structural interventions aiming to foster an enabling environment

An enabling environment underpins all five other objectives in the ICRW framework and is therefore fundamental to interventions aiming to enable adolescent contraceptive use. There were two main activities used in the included interventions to foster an enabling environment: active engagement with communities to change social norms related to gender, adolescence, fertility, and/or contraceptives and activities to demonstrate the value of adolescent girls beyond motherhood.

Several interventions actively and intensively engaged with the community to attempt to develop an enabling environment for adolescent girls’ contraceptive use. For example, in *AGI-K*, committees were established (albeit in the control arm as well as intervention arms) where facilitated ‘community conversations’ were used to “identify key issues in the community that lead to the undervaluing of girls and the perpetuating of violence against girls and women” (p. 3) and to develop an action plan with a small fund to assist its implementation [23].

Activities to demonstrate girls’ value beyond motherhood were rarely explicit and in many cases, it was not clear whether the intervention activities were aiming to or successful in changing perceptions of girls’ value, e.g., schooling, livelihood training, income-generating activities, or cash transfers. Some interventions used innovative activities, for example, *BALIKA* developed girls’ digital skills (e.g., use of computers and tablets, which were novel in the context), which helped change perceptions of girls as liabilities to being “potentially important assets” (p. 36) [21]. In *Berhane Hewan*, married adolescent girls were given seeds, training, and support to start a small garden, which provided food for their families, and were trained in other home-improvement activities, e.g., furniture construction and building fuel-saving stoves [40].

It was rare for an evaluation to assess whether community attitudes or beliefs had changed; one exception was the *GREAT* study, which surveyed adults (as well as adolescents), asking them a range of questions about their attitudes related to gender, adolescent sex, and fertility [35].

## 4. Discussion

We identified a range of structural interventions that aimed to enable adolescent girls’ effective contraceptive use. However, the limitations of the evidence prevented us from identifying which intervention factors were associated with effectiveness. Through a case-based analysis of studies, we propose a mid-range theory for structural adolescent contraception interventions, which recognises that interventions should be tailored to the specific life stage of the adolescent, focus on elements where baseline gaps have been demonstrated, and incorporate intervention activities to ensure that gaps in terms of motivation, agency, and access to contraception are addressed.

### 4.1. Findings from Other Reviews of Structural SRH Interventions

As far as we are aware, this is the most comprehensive review of structural adolescent contraceptive interventions conducted. A review of empowerment and adolescent pregnancy identified nine articles, mostly non-experimental, that covered low-, middle-and high-income countries [42]. The authors included only peer-reviewed papers but recommended that future studies should consider grey literature, which we have done.

Most reviews of structural interventions have focused on other aspects of sexual and reproductive health, particularly HIV prevention [8,43,44,45,46,47], as well as violence prevention [48,49] and child marriage prevention [50]. Reviews that focused on particular types of structural interventions tended to include only a minority or no studies measuring or focusing on contraceptive use [51,52,53,54] or did not focus on adolescents [11,55,56,57] or LMICs [58].

That there are methodological challenges in this field is not news. The need for consistent outcome measures and empowerment indicators [4,55,57,59], issues of the conceptualisations and terminology relating to structural interventions or empowerment [46,53,60] and study design challenges [4,43,61] have all been identified previously.

Others have recognised the value of the ICRW’s framework [1,4]. We are not the first to call for the consideration of different adolescent life stages; others have noted the frequent neglect of some subgroups (e.g., unmarried adolescents) or their differing needs [1,4,10,59,62,63,64]. The importance of contextual variations and the need to understand context have also been recognised [4,43,55,63,64,65,66]. Finally, others have also recognised the importance of an enabling environment [7,59].

However, this paper builds on the existing evidence base in a number of ways. Firstly, it extends 3ie’s Evidence Gap Map by updating and focusing in-depth on a subset of studies within their broad map of adolescent sexual and reproductive health intervention evaluations [67]. Secondly, it further develops the ICRW’s framework, beginning the process of operationalising it and linking it to existing intervention evaluations [9]. Finally, by reviewing the existing body of evidence, we are able to reflect on methodological challenges that prevent the synthesis of impact findings, with the hope that a greater understanding and consensus can be reached between funders, evaluators, and other stakeholders to strengthen the evidence base going forward.

### 4.2. Recommendations for Future Research

We encourage those implementing, researching, and funding activities in this field to engage in discussions around the methodological challenges highlighted. Reaching a consensus around which indicators and outcome measures to use, as well as other aspects of study designs, such as the optimal duration of follow-up (particularly for those targeting very young adolescents), will enhance future studies and their comparisons and syntheses.

Future research is needed to develop our understanding of how interventions can increase agency for contraceptive uptake in adolescents, as well as which aspects of agency are critical for contraceptive empowerment in different contexts.

Further work is now needed to test and potentially refine the proposed theory through the development and evaluation of structural interventions to enable adolescent contraceptive use in a variety of LMIC settings, with a range of adolescent target populations. This is, of course, merely the first step in a process. Beyond selecting intervention activities, work will be needed to explore the feasibility of implementation at scale for a sustained period. We also encourage consideration of the applicability of this framework to high-income countries; it should not be assumed that structural interventions would not be relevant or potentially effective in these settings.

A clear limitation of our study is our inability to ascertain the studies that were ‘likely effective’ or ‘likely ineffective’ in order to identify critical components or features of these groups due to the methodological heterogeneity of the studies. We are limited by the information provided (or unavailable) in the study documentation. Nevertheless, a strength of this study is the comprehensiveness of the search, which identified several unpublished studies. By including a range of intervention types and study designs, we were able to draw on a broader evidence base to develop the proposed theory. A further strength lies in the methods used, which enabled us to consider not only the methods and outcomes reported but also give in-depth consideration to the processes, contexts, and experiences of the studies. Finally, we built on and developed an existing, respected framework, such that we furthered our understanding of how to operationalise it for structural intervention activities.

## 5. Conclusions

A wide variety of structural interventions have been evaluated; however, methodological issues prevent us from identifying the factors associated with effectiveness in terms of increasing contraceptive use in adolescents. We propose a three-step process for developing interventions that enable adolescent contraceptive use in LMICs: first, identify their adolescent sub-population target group so that activities can be tailored to their specific life stage. Second, assess the context to identify what the intervention should focus on. Third, select activities to address these. These steps will aid developers and evaluators of structural adolescent contraceptive interventions to build a stronger evidence base that is of use across a wide range of settings and use scenarios.

## Figures and Tables

**Figure 1 ijerph-19-14414-f001:**
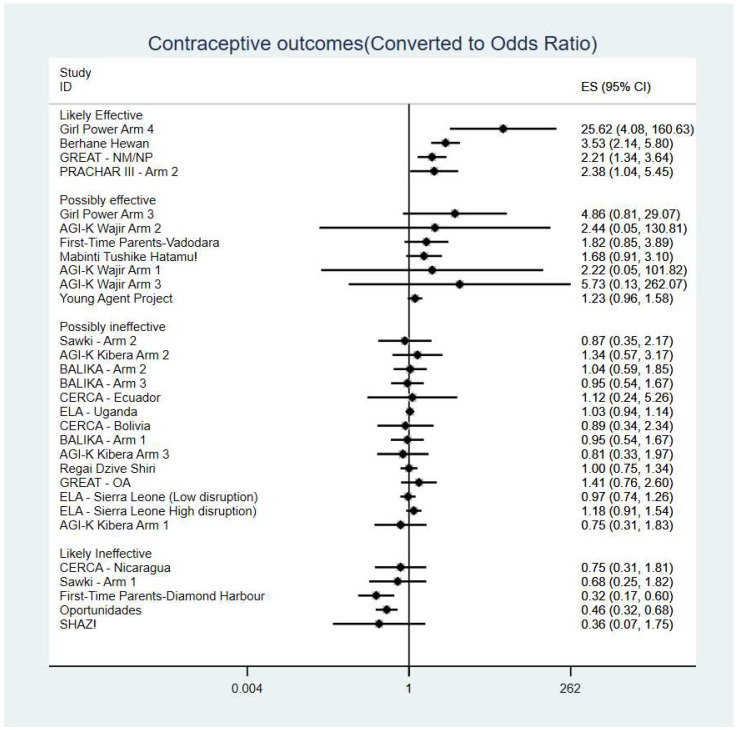
Study arms categorised by likelihood of effect.

**Figure 2 ijerph-19-14414-f002:**
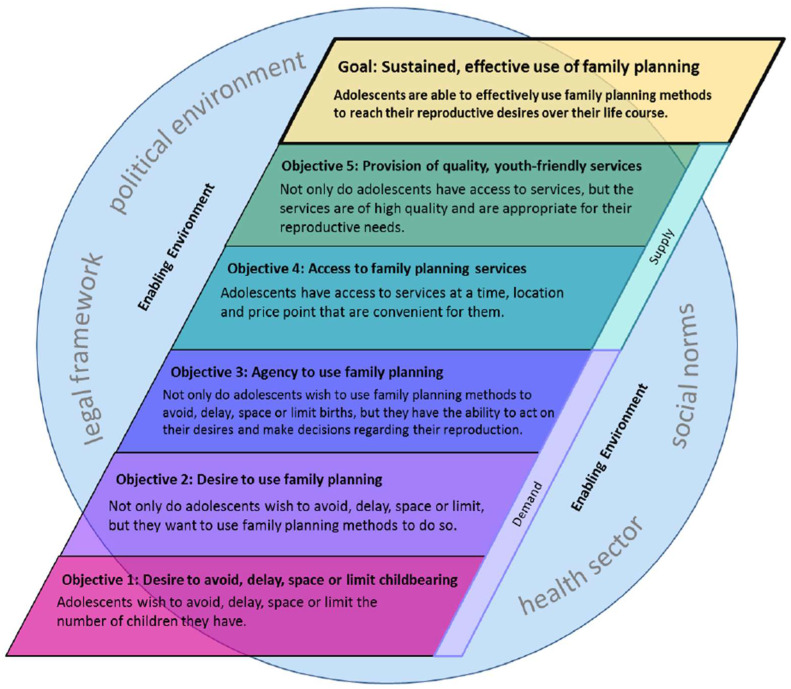
ICRW’s conceptual framework for adolescent family planning use. (Reprinted with permission from ref [9]. Copyright 2014 the International Center for Research on Women).

**Table 1 ijerph-19-14414-t001:** Categories of effectiveness.

Category	Definition
Likely effective	Study intervention arms with an odds ratio (OR) over 1, indicating higher contraceptive use than the control, with a 95% confidence interval (CI) that did not include 1
Possibly effective	Study arms with an odds ratio (OR) of 1.5 or more or with an OR over 1 and a 90% CI that does not include 1 (not 95% CI)
Possibly ineffective	Study arms with an OR between 0.75 and 1.25 and a 90% and 95% CI that includes 1
Likely ineffective or harmful	Study arms with an OR under 1, indicating lower contraceptive use than the control with a 95% confidence interval (CI) that did not include 1, or with an OR lower than 0.75

**Table 2 ijerph-19-14414-t002:** Data table of least/most effective study arms and their methodological characteristics.

Study (Arm)	RCT	Baseline Measurements Available	Control Group	Data Collected from Sexually Active Only	Measure Reflected Current Use	Outcome
Berhane Hewan	No	Yes	Yes	Yes	No	Likely effective
Great (NM/NP)	No	Yes	Yes	Yes	Yes	Likely effective
PRACHAR III (arm 2)	No	No	Yes	Yes	Yes	Likely effective
DISHA	No	Yes	No	Yes	Yes	Likely effective
Girl Power (arm 4)	No	Yes	Yes	Yes	No	Likely effective
Oportunidades	No	No	Yes	No	Yes	Likely ineffective
SHAZ!	Yes	Yes	Yes	No	Yes	Likely ineffective
Sawki (arm 1)	No	No	Yes	Yes	Yes	Likely ineffective
CERCA (Nicaragua)	Yes	Yes	Yes	No	No	Likely ineffective
First time Parents Project (Diamond Harbour)	No	Yes	Yes	Yes	No	Likely ineffective

**Table 3 ijerph-19-14414-t003:** Truth table of selected methodological characteristics.

RCT	Control Group	Data Collected from Sexually Active Respondents Only	Outcome	Number of Studies	Consistency	PRI *
0	**0**	**1**	**1**	**1**	**1**	**1**
0	1	1	0	6	0.667	0.667
1	**1**	**0**	**0**	**2**	**0**	**0**
0	1	0	0	1	0	0

* PRI = proportional reduction in inconsistency.

**Table 4 ijerph-19-14414-t004:** Summary of intervention activities targeting adolescent girls to increase their agency, and example indicators.

Agency Element	Agency Sub-Element	Example Intervention Activities	Examples of Relevant Outcome Indicators
Choice	Aspirations/opportunities	Livelihood experience Support for schooling Employment opportunities	Have hope for their future [25] Preferred number of children [31] Have college aspirations [29]
	Value beyond motherhood	Vocational support Income generating support Practical skills development Employment opportunities	Receive own income [20] Believe that only when a woman gives birth to a child is she a real woman [35] Would you hope to have a job outside of the home even after marriage/having children or would you prefer not to work outside the home if possible? [24]
Voice		Community development/civic engagement projects Communication/negotiation training Sexual/reproductive health training Gender rights training	Whether their family (or spouse) listens when they speak [25] Discuss family planning with spouse [25] Able to talk to spouse about contraception [27] Whether their family (or spouse) considers their concerns when making decisions [25]
Power	Power to make decisions	Decision-making training Experience in decision making Economic empowerment Cash transfers	Whether their family or spouse trusts them with important household tasks [25] Relationship power [20] Able to go to the clinic if I needed to get contraception [19] Believe that their partner would support their decision to use a contraceptive [35] Believe that a man and a woman should decide together on type of contraceptive [35]
	Power within (esteem)	Safe space groups—to build confidence	Are confident they could use contraceptives correctly at all times [35] Felt that they were at least as important as other people [24]
	Power with (support)	Safe space groups—to build a social network Safe space—mentors	Had a friend that she could confide in about spousal relations, sex, pregnancy/childbirth, family planning, wage work, and problems in the marital family [36] Received high social support [20] Permitted to visit friends [21]

## Data Availability

Not applicable.

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
