# Peer review of "Structural Interventions to Enable Adolescent Contraceptive Use in LMICs: A Mid-Range Theory to Support Intervention Development and Evaluation"

_ijerph, 2022, doi:10.3390/ijerph192114414_

Round 1
Reviewer 1 Report
Sexuality is part of life, the body, relationships between people, personal growth, and life in society. Sexuality and affectivity are essential components of intimacy and interpersonal relationships.
Family Planning is an important element of medical care.
It must be directed to young people and adults of both sexes and involves an important dimension: Sex Education. Doctors and nurses must assume this professional responsibility. Why should they? Because the informal and spontaneous sex education that exists always and everywhere is often not sufficient, enlightening and effective. Furthermore, because there is a persistent idea: "to talk about sexuality is to talk about sex and how it can be controlled through family planning". It is a recurrent and wrong idea, not favoring human development at all.
In this sense, sex education should not only focus on contraceptive methods, the issue of abortion, the prevention of sexually transmitted diseases, gender violence or child marriage prevention.
Unfortunately, the text under review does not reflect this comprehensive and humanistic vision of Sex Education and it does not assume that the effective use of contraceptives by girls will be useless if it is not accompanied by a structured and continuous Sex Education over time.
The authors focused their attention on the issue of contraception for adolescent women and forgot everything else.
Author Response
Thank you for your comment – we agree that sexual wellbeing is broader than contraception alone and that sex education should cover not only contraception but other topics as suggested, such as abortion, STIs and gender violence and child marriage.
Unfortunately, our focus on contraception was set by those funding the research.
Since our review was focused on structural interventions, we excluded evaluations of sex education interventions unless they explicitly included a structural component.
Reviewer 2 Report
Thank you for the opportunity to read this interesting submission about this important topic. The messages are spelled clearly and you make a good case for research and funding activities in the field to engage in discussions around the methodological challenges.
I have some comments on the methodology- especially on how the search was conducted but I enjoyed reading this draft.
Comments
The paper is generally well written but there are a few areas that need improving.
1. Search strategy: There was no information on how the search was conducted, what were the search terms, and what were the inclusions/exclusion criteria.
2. It is also not clear which analytical tool was used in analysing the data.
Author Response
Thank you for your positive feedback!
Our search strategy was presented in our previously published article, which we cite here [reference 12]. We have added further clarification here – however if the editors would like us to copy these details across from our other article, we are happy to do this.
We used Intervention Component Analysis – we have made this clearer on page 3.